# 3Rs-Related and Objective Indicators to Help Assess the Culture of Care

**DOI:** 10.3390/ani9110969

**Published:** 2019-11-14

**Authors:** Penny Hawkins, Thomas Bertelsen

**Affiliations:** 1RSPCA Research Animals Department; Wilberforce Way, Southwater, West Sussex RH13 9RS, UK; 2Novo Nordisk A/S, Novo Allé, 2880 Bagsvaerd, Denmark; tsbt@novonordisk.com

**Keywords:** 3Rs, Culture of Care, ethics, research animals, laboratory animal welfare, Animal Welfare Body, replacement, reduction, refinement

## Abstract

**Simple Summary:**

‘Culture of Care’ within animal research and testing refers to a commitment to improve animal welfare, the quality of the science, staff morale, and openness with the public. An effective Culture of Care should also promote the replacement of animal experiments with humane alternatives, reductions in animal numbers and suffering, and better welfare through the refinement of procedures, housing, husbandry and care (collectively known as the 3Rs). The Culture of Care is recognized as the foundation of humane and responsible science, but the concept should be applied in a meaningful way and not simply used as a ‘buzzword’. Recognizing this, some establishments have begun to define and assess their individual Culture of Care. This paper provides some examples of their approaches to surveying staff and external colleagues. It also sets out some suggestions for objective criteria for assessing the Culture of Care, and for indicators that capture progress with each of the 3Rs. The aim is to complement the growing literature on the Culture of Care and highlight some sources of information and inspiration to help establishments tailor their own assessments.

**Abstract:**

Within animal research and testing, the need for an effective Culture of Care is widely recognized and described in terms of an establishment-wide commitment to improving the implementation of the 3Rs, animal welfare, scientific quality, care of the staff, and transparency for all stakeholders, including the public. Ideally, each establishment would determine what the Culture of Care means for them, and be able to assess and potentially benchmark their own culture. Some establishments already do this, using various indicators and formal or informal assessments. This paper provides examples of these approaches to assessing the Culture of Care, including surveys and surrogate measures. Many currently-used criteria and indicators tend to be human-centric and subjective, and we suggest using further objective indicators and animal-centric, 3Rs-based criteria. It is preferable to consider each of the 3Rs separately when assessing culture, and some indicators are suggested to facilitate this. Several documents produced by regulators in the UK and European Union are good sources of objective indicators of a good Culture of Care. This concept paper aims to complement the literature on assessing the Culture of Care, providing ideas and sources of information to help identify relevant and measurable criteria.

## 1. Introduction

In the context of animal research and testing, the term ‘Culture of Care’ is used to describe an establishment-wide commitment to improving animal welfare, scientific quality, care of the staff, and transparency for all stakeholders, including the public [1]. Over the last decade, the need for a Culture of Care has become increasingly widely recognized and is mentioned in good practice guidelines [2,3,4,5]. A working concept for a Culture of Care is available at [1], and several publications provide guidance as to how the Culture of Care can be implemented in practice [5,6,7]. Although the Culture of Care is included in publications and guidelines in a number of countries (e.g., [1,2,4]), the concept is still developing worldwide and is currently mostly included in guidelines and working documents from the European Union (EU). This concept paper therefore uses EU terminology where this is unavoidable, but the approaches to assessing the Culture of Care will apply globally.

It is positive to see that the Culture of Care has become established as an essential foundation for humane and responsible science. However, there is also a concern that it can be used as a ‘buzzword’ (e.g., ‘we have an excellent Culture of Care’) without ensuring that the local Culture is indeed positive, or that all its different aspects are being fulfilled. An establishment cannot be said to have a good Culture of Care if its culture is merely one of compliance, without improving on legal minimum requirements [5,8].

## 2. Assessing the Culture of Care

A genuinely good Culture of Care should lead to benefits for animal welfare, the 3Rs (replacement, reduction, and refinement), the quality of the science, and staff wellbeing. Achieving a Culture of Care is not a goal in itself, but is a means to achieve a goal; the direct and indirect benefits for the animals, as described above. The Culture of Care is a strong and efficient enabler to achieve positive outcomes for animals, humans, and science. This can be envisaged as a house where the Culture is the foundation, the goal (optimal animal welfare) is the roof, and the internal structures (e.g., the Animal Welfare Body; AWB or equivalent committee) are the pillars that support the roof and are firmly based upon the foundation (Figure 1).

Every establishment is different, so each needs to determine what the Culture of Care means for them, and to be able to assess and potentially benchmark their own Culture. Some establishments have begun to do this. For example, a discussion amongst an international Culture of Care network in 2017 found that several establishments had begun to describe and assess their Culture, using surveys which either focused on internal staff perspectives, via questionnaires and discussions, or sought external viewpoints. All were able to list useful indicators, but some had not yet used these within a formal assessment exercise [9].

One participant in the above discussion was the pharmaceutical company Novo Nordisk, which designed and performed an internal Culture of Care survey. The questionnaire was based on the concept that culture can be described as ‘what we do and what we think’, which shapes how individuals and groups interact when working to advance animal welfare. Consequently, the questions were related to the individual (e.g., ‘I feel empowered …’), the group (e.g., ‘in our group we ….’), their relationships with management, and knowledge of the internal supporting structures. Respondents were asked to indicate their role, as it is very likely that there will be role-dependent sub-cultures, especially within a large organization.

The survey used surrogate markers, which were designed to have a clear meaning for a wide audience and to be less likely to be interpreted individually. These surrogate markers were: Collaboration, Trust, Integrity, Influence, Meaning, Predictability, Social Support, Rewards/Recognition and Resources. Thirty-nine questions were included, to be answered using a 5-point qualitative scale (strongly disagree, disagree, don’t know, agree, and strongly agree). Finally, there were optional sections for free text input. The questionnaire was distributed online to 343 potential respondents, of whom 151 replied (a 44% response rate). The quantitative results, together with the free text comments, enabled the AWB and management to identify matters to address at an individual, group, leader, or organizational level. These related to different professional groups, values, and operational issues, and directions were identified as to where and how to initiate potential actions to further the Culture of Care.

This type of approach to assessing the Culture of Care is further developed in [8], and the European Federation of the Pharmaceutical Industry and Associations (EFPIA) Research and Animal Welfare Group (RAW) has also developed a tool to help organizations review and reflect upon their Culture of Care [10].

The above initiatives all represent sound progress with assessing the Culture of Care, often with human-centric criteria and indicators. For example, the survey used by EFPIA RAW includes questions exploring how people feel and what they believe. Typical questions include ‘I feel accountable for animal welfare’, ‘Animal welfare is a priority in your company’, and ‘Our AWB leads a ”Culture of Care”’. These are all good questions with respect to ensuring human welfare and empowerment, which should create an environment that is open to facilitating the 3Rs.

To ensure that assessments encompass the animal welfare and 3Rs benefits of a good Culture of Care, it is also important to evaluate some direct outcomes for animals. This is included in a question within the EFPIA survey: ‘My organization actively promotes the 3Rs in working practices?’ If the respondent agrees, they are prompted to provide examples.

This above approach to assessing the Culture of Care is helpful, and we believe it is especially beneficial to make sure that the evaluation also includes (i) adequate ‘animal-centric’ criteria and (ii) objective indicators, such as project meetings with animal technologists and scientists, 3Rs awards, or an animal welfare focus group. It is reasonable to assume that a culture in which people feel valued (and listened to) will lead to reductions in animal use and suffering, and improvements in welfare, but this cannot be verified without dedicated indicators. From a purist animal advocate’s perspective, one could question the point of a Culture of Care if it does not have a positive impact for animals.

Of course, an overly long list of criteria and indicators would make for a very time-consuming assessment exercise, which would be counterproductive. We have set out some suggestions for indicators below, with the aim of providing additional choices and ideas for establishments when they produce their own tailored assessment exercise to fit their own requirements and resources. Ideally, this would take an integrated approach, incorporating indicators based on management processes, outcomes for animals, and the views of staff.

### 2.1. 3Rs-Based, Animal-Centric Criteria

Each of the 3Rs represents a better outcome for animals, whether this means never having been used at all (replacement and reduction) or experiencing less suffering and better welfare (refinement). The 3Rs are often expressed as a single entity when thinking about the Culture of Care, but always considering all three together may not be helpful when performing an assessment.

Broadly speaking, each of the 3Rs is implemented via different approaches and mechanisms, using different information sources, and each ‘R’ can present different obstacles to overcome. Furthermore, in the authors’ experience, people sometimes refer to implementing the 3Rs when in practice they just mean refinement. This can result in a skewed impression with respect to success with replacing animals, optimizing animal numbers, and improving experimental design.

For these reasons, we suggest that it is better to include each ‘R’ separately when defining indicators. Table 5 within the EFPIA RAW publication includes some useful indicators, and we list some further suggestions for replacement, reduction, and refinement-related indicators in Table 1.

Indicators like these will help to focus on the animals’ experiences, and ask meaningful and relevant questions to help determine how effectively the local culture is promoting (and shaped by) the 3Rs.

### 2.2. Objective Indicators

Assessments of the Culture of Care will also benefit from some measurable criteria that directly indicate both animal and human welfare. Useful documents in this respect are the European Commission Working Documents on Animal Welfare Bodies and National Committees [5], inspections and enforcement [7], and education and training [6]. The first two of these have sections on the Culture of Care, which list key elements that can be used as a basis for objective indicators, e.g., ‘AWB members should receive appropriate individual induction training and Continuing Professional Development’ [5]. Table 2 sets out some examples of indicators taken from the Working Document on inspections and enforcement.

An especially helpful document is the UK regulator’s advisory note on low-level concerns at licensed establishments [11]. Appendix A within this guidance lists indicators of good practice, which correspond closely to the criteria for a good Culture of Care. We have used this document to list some suggestions for objective indicators in Table 3.

Indicators like these can be objectively verified and measured, which means that they can complement the more qualitative, subjective indicators that relate to how people are feeling. An approach that includes appropriate proportions of subjective, objective, and animal-centered indicators should provide a more rounded, integrated analysis of the Culture of Care and how this is developing.

## 3. Conclusions

The authors hope that this concept paper will complement the literature on assessing the Culture of Care, by providing ideas and highlighting sources of information to help ensure that indicators are relevant and that at least some are measurable. The key take-home messages are:Make sure that there are some separate indicators relating to (i) replacement, (ii) refinement, and (iii) reduction, including optimizing experimental design and analysisThink about the life-time experience of the animals, how a good Culture of Care should improve this, and how this might translate into indicators you could assessIdentify some objective indicators (such as animal welfare outcomes, supportive structures, or cultural descriptors) that can be measured or evidenced, e.g., with audit trailsReview references 3 to 11 for inspiration, and look at the Norecopa Culture of Care pageShare your assessment protocols, e.g., on your establishment’s website and via articles and blogs, and consider joining relevant discussion groups such as the Culture of Care network, or networks for AWBs, ethics committees, or Institutional Animal Care and Use Committees

## Figures and Tables

**Figure 1 animals-09-00969-f001:**
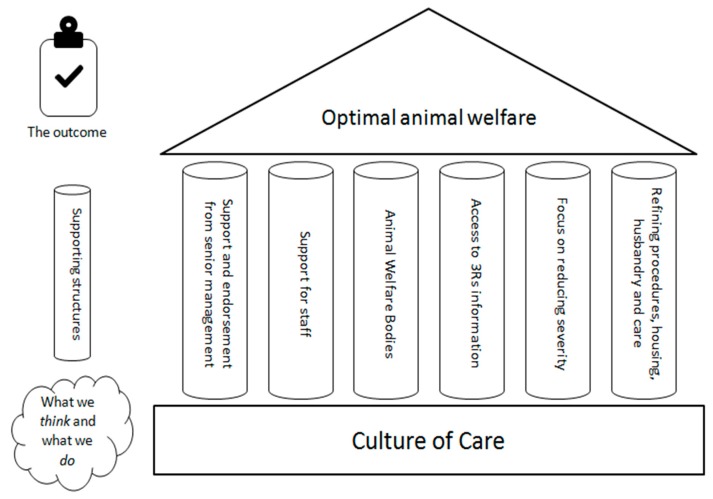
Schematic showing how the Culture of Care underpins good practice and optimal animal welfare.

**Table 1 animals-09-00969-t001:** Suggestions for indicators of success with respect to implementing the 3Rs.

All 3Rs	Can staff demonstrate appropriate knowledge and awareness of the 3Rs, including the ability to define them correctly and knowing where to access information and guidance about each one? What access do people have to information about replacement, refinement, and optimising animal numbers; e.g., databases, journals? Are staff supported in attending 3Rs-related meetings?
Replacement	Do researchers actively and regularly search for potential replacements using relevant databases, and can they demonstrate this? Are animals replaced as far as possible within staff training? Do meetings take place between those working in different disciplines, to share ideas and approaches to replacing animals?
Reduction/optimising animal numbers	Do researchers have access to someone with expertise in biostatistics, and is this person consulted? Is there a strategy to minimise animal surplus due to overbreeding, and how effective is this? Is there a strategy for sharing tissues, and does it work well? Is there a programme to share and archive genetically altered lines?
Refinement	Data on the fates of animals, e.g., how many were humanely killed as part of the experiment; euthanased because a humane endpoint has been implemented; found dead; died under anaesthetic; used for tissues (as above); or rehomed? Provision of environmental enrichment; what do animals have in their enclosures; is this good practice; how frequently is this reviewed; do animals with special needs have appropriate enrichment; if animals do not have standard enrichment, is this justifiable? Are humane killing methods regularly reviewed, instead of defaulting to the easiest method that is permissible under the legislation? Is good practice observed with respect to asepsis? What are trends in actual severity within projects and how are these explained? Are humane endpoints regularly reviewed and refined?

**Table 2 animals-09-00969-t002:** Examples of criteria, reproduced from the European Commission Working Document on inspections and enforcement [7], which could be adapted into indicators to assess the Culture of Care.

Could Be Good or Bad	Good Culture of Care	Poor Culture of Care
Condition and care of animals Quality of project documentation First impressions such as on state (condition and tidiness) of support areas e.g., the cage washrooms (hardest work—respect to all levels) Status including formal authority of key people—empowerment of staff Attitude of researchers towards the establishment AWB Knowledge of staff on their responsibilities Level of openness of staff and willingness to draw attention to problems	Openness of all staff: keen and able to answer questions Effective designated veterinarian whose input is respected by researchers and care staff High quality, respected care staff On-going education and training in animal care and welfare which is accessible to and encouraged for all levels of staff Effective communication between care staff and research workers e.g., regular meetings; experimental planning Engagement with animal welfare science community, use of biostatisticians Well-understood and clear procedure for ‘whistle-blowing’	Project leader being too distant or removed from research workers and care staff Status of staff—not encouraged to contribute or listened to Care staff/junior researchers not aware of the project details e.g., regarding care, management of adverse effects Resistance to change/introduction of refinement and improvements Lack of understanding of/poor engagement with animal welfare issues by scientists Poor communication between scientists and care staff

**Table 3 animals-09-00969-t003:** Suggestions for objective indicators of a good Culture of Care, based on [11].

Staff numbers appropriate to the size of the establishment, type of work, and type of animals Low turnover of staff and minimal need for agency staff to ‘fill the gaps’ Staff have sufficient time and resource for daily, adequate routine monitoring Attending veterinarian visits regularly and is sufficiently available to provide advice Person responsible for overseeing the welfare and care of the animals (e.g., Named Animal Care and Welfare Officer in the UK, Directive Art 24 1a) meets regularly with users and is aware of their work Person responsible for ensuring that staff have access to species-specific information (e.g., Named Information Officer in the UK, Directive Art 24 1b) has adequate resource for the role Person responsible for ensuring compliance (e.g., Establishment Licence Holder in the UK, Directive Art 20 2) regularly meets with the other responsible persons and the Animal Welfare Body Clear audit trails of communications between scientists and animal technologists (see also examples from the Culture of Care network [12]) A clear system for raising concerns that is supported by management Well-maintained training records Program to review and reassess competence Regular Animal Welfare Body meetings with feedback to staff

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
