# Peer review of "3Rs-Related and Objective Indicators to Help Assess the Culture of Care"

_animals, 2019, doi:10.3390/ani9110969_

Round 1

Reviewer 1 Report

This paper aims to complement existing literature on Culture of Care in animal experimentation and provide suggestions and inspiration for establishments which wish to assess their own Culture of Care. The paper is overall well structured and clear but there is room for improvement in order to make this paper a really instrumental contribution to the existing literature:

1) The authors start with an example of assessing the Culture of Care within an establishment (Nova Nordisk) that is very focused on what the personnel experience, and argue that this is relevant but needs to be complemented with evaluation of direct outcomes for animals. I don’t think this line of argument comes across very clearly in the paper, and I also don’t think that it is conceptually correct. To focus on direct outcomes for animals would include measuring things such as changes in animal numbers and project severity. These are of course relevant measures as they are the ultimate proof that practice has the desired result. But I would argue that in the specific context, the most appropriate measures are of what actual human practice the Culture of Care translates into. Which are also the type of measures you propose. In my understanding, it would be more correct to say that the Nova Nordisk assessment focuses on the Culture, whereas what you propose is a focus on explicit measures of the Care.  

2) Although the concept is mainly discussed in Europe, Culture of Care is relevant for any establishment in any part of the world. By referring to concepts that are specific to a region or a country, the paper comes across as unnecessarily Eurocentric. Examples: Animal Welfare Body (throughout), national legislation (Table 3).

3) Table 3 needs some more work for coherence and completeness. If you aim to make this paper a really practical contribution to the literature – and I think that’s a reasonable ambition! – you need to make sure that this list really includes all the most relevant issues. Under refinement, there is almost nothing on procedures. In the context of culture, one very relevant consideration is how new researchers are trained and how knowledge is transferred! Also, I’m sceptical regarding the relevance of database searches for replacement in research. This presumes that the main issue is to overcome lack of knowledge, and that finding information about an alternative is important for change. This is not my experience from academic research, where there is no standard for what is considered an equivalent alternative and the demands for what would be acceptable are high. Here I see two main sets of obstacles. One has to do with conservative thinking. This has effects well beyond the conservative individual’s own practice as it also leads to others being afraid that their research will not be well received by reviewers if they leave out the in-vivo step. The other has to do with the fact that the really promising alternatives to in-vivo research models are highly complex and often expensive technology, which requires resources in terms of funding, infrastructure and support which may not always be available. I think a Culture of Care which is sensitive to replacement needs to somehow tackle these two. Finally, I recommend that you use a more uniform way of presenting measures here – either statements or questions, not a mix of the two.

Minor issues:

Line 71 Here you refer to a “survey respondent” but I don’t think you have previously explained that there was a survey?

Line 85 The reader needs to know what Research and Development is and what kind of personnel works there

Lines 90-93 Please explain briefly what is in these tools

Lines 107-8 A culture in which people feel valued will not automatically lead to improvements in the 3Rs

Lines 116-7 Whereas this argument seems convincing at a first glance, in reality the animal that is not to be used in research will most likely never be born!

Anna Olsson

Author Response

We thank the reviewer for her comments. We believe that we made the argument for complementing with direct outcome measures in the paper, especially in lines 104-110. We also included project severity in Table 1. We don’t agree that changes in animal numbers are an appropriate indicator of the Culture of Care though, as they are not a metric of the 3Rs either (optimising animal numbers can mean using more, and refinement can mean increasing numbers). We don’t agree that ‘Culture’ and ‘Care’ can be separated, and we agree with the concept that the Culture of Care is defined as ‘a commitment to improving animal welfare, scientific quality, care of the staff and transparency for the stakeholders’ (see norecopa.no/coc).

We agree that the example from Novo Nordisk does not address direct outcomes for animals. We think it is relevant to distinguish between ‘what does the culture look like’ and ‘how efficient is the culture in term of outcomes’. The first is describing ‘qualities’ like values, behaviour and relations – and that is what the referred example is addressing – whereas the other describes tangible outcomes that typically can be measured by using Key Performance Indicators or similar tools. To our knowledge there are no publications describing the latter.

We agree that the paper should be less Eurocentric and have addressed this in lines 45-49. We have also added ‘or equivalent committee’ to AWB in line 63 and our final bullet point at line 174 also mentions other bodies internationally.

We note the comments about the tables. In the paper, each one has a different aim. Table 1 aims to complement the forthcoming EFPIA RAW paper with some further suggestions for 3Rs-related criteria, Table 2 aims to direct the reader to the EC Working Document on inspections and enforcement, and Table 3 aims to direct them to the advisory note on low-level concerns. For Tables 2 and 3, this reflects our aim to ‘complement the growing literature on the Culture of Care and highlight some sources of information and inspiration to help establishments tailor their own assessments’.

There is much more on refinements in the EFPIA paper, as we mention immediately above Table 1. We thought it was very important to complement, and not repeat, other publications. Training and knowledge transfer is included in Table 2. The point about databases is well made, but we still feel that these are useful as surrogate indicators; if the culture is such that people are motivated to search for replacements, and know where to look, they will do so – and if they are working in a regulatory environment, this will be much more relevant, We do emphasise that these are suggestions that can be considered for a tailored approach.

We have altered the indicators so that they are all questions.

Line 71 – thank you, we have clarified this in line 75.

Line 85 – in the interests of brevity, it is easier just to delete (not essential information).

Lines 90-93 – again, we especially wanted this to be a very short paper that would complement the literature, so would prefer not to go into too much detail.

Lines 107-108 – we only said this was a ‘reasonable assumption’, not a given.

Line 116-117 – yes, that is precisely what we meant!

Reviewer 2 Report

In the context of the timescale of the creation and implementation of the 3Rs the Culture of Care is a relatively more recent concept, still under development and arguably more focused on northern Europe and northern America. In additional it has in its earlier iterations had more of a focus on internal process and culture. External communication, audit and opens is a more recent expression of the concept. So a discussion on how its impact can be assessed is important for the development of the concept of the Culture of Care.   This article is for an international journal, therefore I would question whether Culture of Care is recognised internationally as 'a foundation of humane of responsible science', especially when this is in the context of a special issue edition on the 3Rs. Thought needs to be given to placing the history and current status of Culture of Care in the context of the wider worldwide implementation and status of the 3Rs. For example, whilst ‘Climate of Care’ is cited as a precursor for Culture of Care from 2010, earlier work was already using Culture of Care, which an more internal focus: Klein HJ, Bayne KA. 2007. Establishing a culture of care, conscience, and responsibility: Addressing the improvement of scientific discovery and animal welfare through science- based performance standards. ILAR J 48:3-11.   I am not convinced the statement at Lines 49-51 stands up to scrutiny.  Figure 1 could easily represent the current iteration of the EU legislation on use of animals in science. Is meeting the requirements of this legislation, with its iterative review processes, by the authors definition therefore sub-optimal for the 3Rs?    Culture of Care is clearly rapidly advancing as a concept with parts of Europe, but it is unfortunate for critical review that References 7 and 9 are not available as ‘in press’.   There is an inconsistency in approaches between the statement in Lines 109-110 and Table 3, the former looking to outcomes for animals, the latter a list for management processes. As below is it more about integrating both approaches?   Overall, the authors openly highlight this, as in Line 156, as a concept paper, although this rather conflicts with the more definitive statement in Lines 12-14.     - What is be delivered by the 3Rs is clearly very dependent on the underlying people process and the cultural motivation of those people. Reflecting these internal aspects externally, and objective measure, is clearly essential for the social licensing to use animal in science in many parts of the world. But this is work in progress around the world.   -  A wider international perspective, extending  to Asia and elsewhere, more recognition that Culture of Care is a still developing concept, and smarter integration of how animal welfare outputs are linked in measurement of people aspects by the Culture of Care would improve this paper and the help uptake of this important approach.

Author Response

We thank the reviewer for their comments. We have included the reference they suggested (now [2]) and clarified the global applicability of the paper in lines 45 to 49.

The statement in lines 49-51 is substantiated by the EC Working Document on Animal Welfare Bodies, which says: ‘Simply having animal facilities and resources which meet the requirements of the legislation will not ensure that appropriate animal welfare, care and use practices will automatically follow.’ We have added this reference to the statement, and we believe this also addresses the reviewer’s comments about Fig. 1.

Re lines 109-110 and table 3; yes, it is about integrating both approaches as appropriate for each individual establishment. We have further emphasised this in the sentence now at lines 120-121 and by adding ‘also’ to line 109.

Reviewer 3 Report

The primary purpose of this paper is to identify and promote “objective criteria for assessing the Culture of Care”. My very simple point is that not one of the objective criteria proposed is in fact measuring the Culture of Care. As the authors state themselves “From a purist animal advocate’s perspective one could question the point of a Culture of Care if it does not have a positive impact for animals.” This is quite an odd statement, as surely a Culture of Care for animals is the positive impact for animals? Culture of care is the treatment of animals. There is not a Good Culture of Care and how the animal is treated; they are one and the same. The treatment of the animal is the manifestation of the Culture. You simply cannot have a Good Culture of Care and poor care! The objective criteria that the authors identify are entirely focused on whether certain processes are present or absent in different institutions e.g. “Effective communication between care staff and research workers e.g. regular meetings; experimental planning.” It may be more likely that a good Culture of Care might be present when such procedures are in place, but they are emphatically not an objective measure of a Good Culture of Care. It is also worth noting that the some of the criteria suggested as objective are in fact highly subjective e.g. “High quality, respected care staff.” Measuring the respect that is given to staff is incredibly problematic and is not an objective criterion give the difficulty in operationalising the term. The term objective is itself extremely slippery (see Daston, 1992). It would also be possible to have all the indicators of what are identified as a “Good Culture of Care” and the treatment of animals in fact to be very poor (and vice versa). What the authors have identified are processes and systems that it seems reasonable may help promote a Good Culture of Care. However, no evidence is presented at all that these systems and processes are in fact associated with a Good Culture of Care.

I think all the suggestions made by the authors about what to encourage in institutions are all really sensible. However, I think the authors need to be hugely more circumspect in their claim that these processes are objective measures of a culture of care. The only real measures of a Culture of Care would involve a direct examination of how animals were being treated and this may involve qualitative/subjective judgements. I do not regard qualitative judgements as inherently inferior to “objective” measures because seeming objective measures are often ultimately based on qualitative judgements.

Daston, L. (1992). Objectivity and the escape from perspective. Social Studies of Science, 22, 597-618.

Author Response

We thank the reviewer for their comments, but are unclear as to what they would like us to do. We did not actually refer to directly ‘measuring’ the Culture of Care, because we do not believe that this is possible. We discussed surrogate measures and measurable indicators/criteria, but these are not the same as ‘measuring’ a Culture. For example, in all the tables we were very careful to present their contents as suggestions for indicators of a good Culture of Care. We did not claim they were objective measures. Nor did we imply that qualitative judgements were inferior; we made the case for a balanced approach to assessment that involved both examining people’s feelings and attempting more objective (as opposed to subjective) indicators and outcomes for animals.

We also disagree that ‘Culture of Care is the treatment of animals’, when thinking within the scientific community has moved on, such that the Culture of Care is defined as ‘a commitment to improving animal welfare, scientific quality, care of the staff and transparency for the stakeholders’ (see norecopa.no/coc).

We take the point that we are unable to present evidence that the systems and processes are associated with a good Culture of Care. This is because assessing the culture at establishments is at a very early stage, with many (probably most) not yet doing this at all. There is growing interest in, and recognition of the importance of, assessing culture, and this paper aims to consolidate the literature and help establishments to make meaningful progress. That is also why we chose to submit the manuscript as a Concept Paper.

Round 2

Reviewer 2 Report

Thank you for the revision  and I recommend acceptance.

Reviewer 3 Report

In their response to my initial review the authors pointed out that the "We discussed surrogate measures and measurable indicators/criteria, but these are not the same as ‘measuring’ a Culture". I could not agree more - but the truth is that in regard to attitudes, emotions, culture etc. we never have perfect direct measures; we only ever have indirect measures or indicators of any sociological or psychological phenomenon. I think from the title most people would think they are claiming to provide indicators of a culture of care. However, my fundamental point is I just don't understand why the authors are wrapping up their eminently sensible suggestions about good practice in institutions conducting animal research in the discourse of a culture of care. As a checklist of good practice the suggestions are great of themselves. Raymond Williams in his book Keywords (1976) identified 'culture' as just about the most problematic word in the entire language. I think that the authors should simply make the case for their check list as indicating best practice. From an academic perspective, I just think it is entirely pointless to place them in the discourse of culture of care. I do recognise that 'culture of care' is a fashionable phrase at the moment and perhaps any suggestions regarding good practice have to be wrapped up in such a discourse to get any traction. But for me the phrase 'culture of care' is entirely vacuous. My disagreement with the authors is philosophical rather than a criticism of quality. In my view it is not for reviewers to try to insist that authors adopt the perspective of the reviewer. I am more than happy for the article to be published without change if I failed to persuade the authors of the merit of my perspective.